# Immunogenicity and Efficacy of Vaccination in People Living with Human Immunodeficiency Virus

**DOI:** 10.3390/v15091844

**Published:** 2023-08-30

**Authors:** Eeva Tortellini, Yann Collins Fosso Ngangue, Federica Dominelli, Mariasilvia Guardiani, Carmen Falvino, Fabio Mengoni, Anna Carraro, Raffaella Marocco, Patrizia Pasculli, Claudio Maria Mastroianni, Maria Rosa Ciardi, Miriam Lichtner, Maria Antonella Zingaropoli

**Affiliations:** 1Department of Public Health and Infectious Diseases, Sapienza University of Rome, 00185 Rome, Italy; yanncollins.fosso@uniroma1.it (Y.C.F.N.); federica.dominelli@uniroma1.it (F.D.); mariasilvia.guardiani@uniroma1.it (M.G.); carmen.falvino@uniroma1.it (C.F.); fabio.mengoni@uniroma1.it (F.M.); anna.carraro@uniroma1.it (A.C.); patrizia.pasculli@uniroma1.it (P.P.); claudio.mastroianni@uniroma1.it (C.M.M.); maria.ciardi@uniroma1.it (M.R.C.); mariaantonella.zingaropoli@uniroma1.it (M.A.Z.); 2Infectious Diseases Unit, SM Goretti Hospital, Sapienza University of Rome, 00185 Latina, Italy; raffaella.marocco@uniroma1.it (R.M.); miriam.lichtner@uniroma1.it (M.L.); 3Department of Neurosciences, Mental Health, and Sense Organs, NESMOS, Sapienza University of Rome, 00185 Rome, Italy

**Keywords:** HIV, PLWH, ART, vaccination, immune responses, CD4, COVID-19, HPV, influenza

## Abstract

People living with HIV (PLWH) remain at high risk of mortality and morbidity from vaccine-preventable diseases, even though antiretroviral therapy (ART) has restored life expectancy and general well-being. When, which, and how many doses of vaccine should be administered over the lifetime of PLWH are questions that have become clinically relevant. Immune responses to most vaccines are known to be impaired in PLWH. Effective control of viremia with ART and restored CD4+ T-cell count are correlated with an improvement in responsiveness to routine vaccines. However, the presence of immune alterations, comorbidities and co-infections may alter it. In this article, we provide a comprehensive review of the literature on immune responses to different vaccines in the setting of HIV infection, emphasizing the potential effect of HIV-related factors and presence of comorbidities in modulating such responses. A better understanding of these issues will help guide vaccination and prevention strategies for PLWH.

## 1. Introduction

In people living with HIV (PLWH), reduced immune responses to most vaccines are known [1,2]. Antiretroviral therapy (ART) restores life expectancy and general well-being, reducing the risk of severe outcomes after infection in PLWH [3,4].

HIV infection induces a profound disruption of both the innate and adaptive immune systems leading to immunological alterations and persistent immune dysfunction [5]. Primary infection elicits systemic immune activation and inflammation followed by a progressive loss in CD4+ T-cell count and a persistent expansion of circulating CD8+ T cells [6]. Furthermore, exhaustion of T cells often recurs, together with an alteration of the innate immune cell functions [6,7]. Indeed, alterations of B-cell activity such as abnormal activation and lower antibody responses have been described [8].

ART-induced suppression of HIV replication is associated with a significant increase in absolute CD4+ T-cell and B-cell counts, including naïve and memory cells that are essential for humoral and cellular immunity to T-cell-dependent and independent immunogens [9]. However, despite effective virological suppression, chronic activation persists and antigen-specific T- and B-cell responses, including T follicular helper cell (Tfh) functions, are still impaired. Furthermore, PLWH continue to have higher levels of inflammatory mediators, such as interleukin (IL)-6, tumor necrosis factor-alpha (TNF-a), soluble (s) CD163, sCD14 and C-reactive protein (CRP)-accelerated aging, and some other comorbidities may accompany this therapy [10,11]. Some molecules with immunomodulatory properties have been shown to have some beneficial effects on this residual inflammation [12].

However, in PLWH, this impairment of the immune system may affects the quantity, quality and persistence of protective immune responses induced by natural infection or vaccination, reducing responsiveness to vaccines and their effectiveness [13,14,15]. In addition, vaccine-induced antibodies may decline more rapidly than in the general population [16].

As reported in Table 1, guidelines recommend a proactive approach for immunizing PLWH who are susceptible to vaccine-preventable infections and at risk of exposure, including those who have received previously contraindicated live attenuated vaccines such as those against measles, mumps and rubella [17]. On the contrary, the bacillus Calmette–Guérin (BCG) continues to be contraindicated in PLWH due to its unfavorable benefit/risk profile [18,19].

In general, the effective viremia control by ART and the improvement in the absolute CD4+ T-cell counts are correlated with an enhancement in responsiveness to routine vaccines, although this issue continues to be of concern. Together with current CD4+ T-cell absolute count, the CD4/CD8 ratio has proved to be an accurate predictor of vaccine success [20] (Figure 1).

Finally, co-infections represent an additional factor that may influence immune responses to vaccination, due to their contribution to a persistent immune activation state and induction of immune senescence [21,22].

Overall, vaccination of PLWH remains challenging and with the present review, we summarized recent works in the literature on different vaccine responses in the setting of HIV infection (Table 1).

## 2. Hepatitis A Virus Vaccination

Hepatitis A is a viral infection caused by the Hepatitis A virus (HAV), a single-stranded RNA virus from the *Picornaviridae* family [23]. HAV is commonly transmitted through the fecal–oral route (through ingestion of contaminated food or water), person-to-person contact and men who have sex with men (MSM) [24,25] and may be responsible for forms of acute hepatitis that may progress to fulminant hepatic failure in non-immune adult populations [26,27].

Hepatitis A occurs worldwide and is highly endemic in the most precarious areas of low-income countries [28,29]. It was estimated to have caused, in 2005, apart from approximately 200 million subclinical and oligo-symptomatic HAV infections, 33 million cases of symptomatic illness and 35,000 deaths [30,31].

In Europe, the HAV seroprevalence is low and observational data suggest that PLWH, especially MSM and injecting drug users (IDUs), are at increased risk of contracting HAV [24,32,33,34]. Additionally, a small study conducted on 15 PLWH with acute hepatitis A showed that the duration of HAV viremia was prolonged compared to the general population with acute hepatitis A, which may increase the likelihood of contracting HAV and transmission to others [35]. Overall, older age, IDUs and MSM have been identified as independent factors associated with HAV seropositivity in PLWH [36,37,38,39].

Among the preventive measures to reduce the spread of hepatitis A, vaccination against HAV remains the most effective [40,41]. Currently, two types of HAV vaccines are available: the live attenuated vaccine and the inactivated HAV vaccine [42]. Only the latter is recommended for PLWH [43] including different adult, adolescent and pediatric formulations [43].

In the literature, several studies have evaluated the effectiveness of vaccination against HAV in PLWH. A low rate of seroconversion compared to the general population was observed. Absolute CD4+ T-cell count < 200 cells/µL, viral load, old age, CD4/CD8 ratio, hepatitis C co-infection and gender were identified as factors of poor response after vaccination [44,45,46,47].

Several previous studies evaluating the efficacy of HAV vaccination in PLWH have shown lower seroconversion rates (vaccine efficacy) than their seronegative counterparts [48,49,50,51,52,53]. However, a recent prospective observational study in the setting of an epidemic of acute hepatitis A among MSM in Taiwan observed an overall seroconversion rate among PLWH MSM of 39.7% and 93.4% after receiving one dose and completing a two-dose series of HAV vaccination, respectively, and despite the delayed serological response, HAV vaccination resulted in a 93% reduction in the risk of acute HAV infection in HIV-positive MSM over the course of the epidemic [43]. Higher absolute CD4+ T-cell counts were consistently correlated with higher seroconversion rates [43].

In another study evaluating the efficacy of the vaccine against HAV in a group of 29 children, including 6 children living with HIV and having lost their HAV seropositivity 7 years after being vaccinated with an inactivated vaccine against HAV; after revaccination (two doses), 83% of these PLWH had a seroconversion after the first dose [54].

Other studies evaluating the efficacy of the HAV vaccine according to the increase in the number of vaccine doses have shown that increasing the number of doses from two to three increases the rate of seroconversion in PLWH [52,53,55,56,57].

Various studies evaluating the persistence of immune memory have demonstrated that in healthy adults following a primary two-dose regimen, anti-HAV antibodies can persist in >90% of vaccines for 40 years or more [58]. In PLWH, on the other hand, a slight decrease was observed over time. In other prospective studies investigating PLWH (all with an inactivated vaccine against HAV), better results of the persistence of immune memory were observed with a duration of seroprotection in PLWH patients equivalent to 7 years in 79% of 29 adolescents, 94 for 6–10 years in 85% of 116 adults, 117 for 3.7 years in 85% of 52 adults [59], and for 5 years in 75.5% of 49 adults [47].

Different studies assessing the persistence of immune memory have observed that in healthy adults following a primary two-dose regimen, anti-HAV antibodies can persist in >90% of vaccines for 40 years or more [58]. In prospective studies of successfully vaccinated PLWH (all with an inactivated HAV vaccine), there was a slight increase in the persistence of seroprotection, which in some of these PLWH (both adults and adolescents) oscillated between 5 and 10 years old [47,54,59,60].

The persistence of immune memory was also confirmed by a recent study comparing three-dose and two-dose HAV vaccination schedules, where a slightly higher seroprotection rate of 94% versus 88% was found after 5 years in 155 and 95 adults, respectively [61].

Despite the evidence of efficacy conferred by the HAV vaccine, PLWH remain susceptible to HAV infection in high-income countries, due to low compliance with recommended HAV vaccination guidelines, at-risk sexual behaviors and injecting drug use.

## 3. Hepatitis B Virus Vaccination

Hepatitis B is a liver viral infection caused by the Hepatitis B virus (HBV), a DNA virus belonging to the *Hepadnaviridae* family [62].

Because HIV and HBV share similar routes of transmission, co-infection with the two viruses is common [63,64]. HBV infection in PLWH is a global public health problem [65]. Globally, nearly 10% of PLWH are infected with HBV [66,67]. HBV infection in PLWH is generally characterized by an increased rate of cirrhosis (10–20%), a higher risk of hepatocellular carcinoma [63,68,69,70] and lastly a higher risk of liver-related death [70].

In general, HBV is not cytopathic. It causes damage through the induction of immune mechanisms. Cytotoxic CD8 cells recognize expressed HBV antigens and destroy infected hepatocytes, resulting in increased aminotransferases [71]. Thus, HBV establishes a persistent infection with a stable reservoir of genetic material in the form of circularized DNA in the cell nucleus [65]. The different phases of HBV infection are characterized by the presence of certain viral and immunological markers allowing the orientation of the therapeutic decision and the evaluation of the response to treatment. Surface antigen (HBsAg) is the first marker detected in serum [72]. Its presence indicates HBV infection [73]. The disappearance of HBsAg is followed by the appearance of anti-HBs antibodies. Anti-HBs is considered as a neutralizing antibody and is recognized as a marker of disease protection and cure [73]. In most patients, anti-HBs persists for life, confirming long-term immunity. Anti-HBs is only serological marker in individuals who have an immune response after vaccination against hepatitis B [74].

HBV vaccination is recommended in PLWH as the most important method of prevention [75]. Despite these recommendations in 2015, only 2/3 of PLWH receive at least one dose of HBV vaccine [75]. The efficacy rate of this vaccine in terms of immune response is generally defined by seroconversion with anti-HBs antibodies > 10 µL/mL in these subjects [76].

The first studies on the efficacy of the vaccine against HBV applying the “classical” schedule (20 µg of HBs antigen at months 0–1–6) showed relatively low seroconversion rates in PLWH, with only 20–70% overprotected against 90–95% in the general population [77,78]. Nevertheless, in these studies, a low rate of response to vaccination could be correlated with various risk predictors of poor response, including viral load and absolute CD4+ T-cell count, CD4/CD8 ratio, co-infection with HCV, poor general health and occult hepatitis B [79,80,81,82,83]. In addition, the female sex [84,85], younger age [86,87,88], alcohol consumption [86] and smoking [88] are all factors of negative response to vaccination. 

Recent studies have shown improvements in the effectiveness of the HBV vaccine in PLWH, particularly with seroconversion rates [89,90,91]. In Uganda, to assess the efficacy of HBV vaccines in PLWH, a cohort of 132 participants received both the base dose of the vaccine and the 1-month dose, and 127 received the 6-month dose. The 132 participants who entered the study were predominantly female and 52% had received ART for ≥3 months and 94% had undetectable HIV RNA. The median (IQR) CD4+ T-cell count was 426 cells/μL (261–583). A high humoral response rate in PLWH was seen. Nevertheless, in this study, a variation was observed in the immune response of these PLWH to the vaccine, with 86% of participants that were high-level responders with anti-HBs titer levels ≥ 100 IU/L, while 6% were low-level responders (anti-HBs levels 10 to 99 IU/L) and 10 (8%) were non-responders (titer levels < 10 IU/L) [89].

This high response rate differs from some studies that have shown suboptimal seroconversion rates in response to the standard series of HBV vaccines in PLWH. These results are like a study in China that also found high response rates to the HBV vaccine in PLWH. Nevertheless, compared to the general population in China [90], the response to HB vaccination was diminished in PLWH.

In a study in Thailand, high rates of response to the HBV vaccine were also observed in PLWH with fully suppressed HIV viral load and absolute CD4+ T-cell count ≥ 200 cells/µL [88].

Given the importance of immune status in vaccine response, it is possible that participants’ degree of immune reconstitution differed between studies, despite apparently similar current absolute CD4+ T-cell counts documented in these studies. On the other hand, in the literature, we find results of the efficacy of the vaccine against HBV, correlated with the increase in vaccination schedules against HBV. Launay et al. [92], in one study, found that PLWH vaccinated with a four-double-dose schedule had higher anti-HBV titers and stronger immune responses than those vaccinated with the standard three-dose schedule (82% versus 65%, *p* < 0.05). Chaiklang et al. compared the immunogenicity and safety of three standard doses and four double doses versus four standard doses in one RCT [88]. A large randomized trial evaluating three-dose 20 μg and four-dose 40 μg HB vaccination regimens in PLWH reported seroconversion rates of 65 and 82%, respectively [92]. Prospective studies and randomized trials have reported similar response rates of 50–62% [93,94], while other studies have reported similar or better rates, ranging from 84 to 92% [88,89,95]. However, these studies with higher seroconversion rates exclusively enrolled patients with absolute CD4+ T-cell counts of 200 cells/µL or higher or included patients regardless of whether they had received an HBV vaccine in the past. 

In another study evaluating the persistence of vaccination, O’Bryan et al. [96] followed 186 HIV patients for 5 years and found that the persistence of the HBV vaccine response was longer when these patients had an undetectable or low viral load [97].

These data demonstrate an improvement in the efficacy conferred by the HBV vaccine, but achieving a long-lasting and protective level of immunity remains a challenge in patients with detectable HIV RNA or low CD4+ T-cell counts at the time of vaccination.

## 4. Human Papillomavirus Vaccination

Human papillomavirus (HPV) infection represents the most prevalent sexually transmitted infection in the world [98]. HPV is a small, non-enveloped DNA virus infecting skin or mucosal cells that belongs to the *Papillomaviridae* family. The genome encodes for six early proteins responsible for virus replication and two late proteins, L1 and L2, which are the viral structural proteins [99]. At least 13 of more than 100 known HPV genotypes can cause cancer of the cervix and are associated with other anogenital cancers and cancers of the head and neck [99]. HPV types are divided into high risk, associated with the development of anogenital cancer, and low risk, rarely associated with the development of cancers [100]. The two most common “high-risk” genotypes (HPV 16 and 18) cause approximately 70% of all cervical cancers [99].

In immunocompetent individuals, most HPV infections spontaneously resolve; however, the persistent infection with oncogenic HPV genotypes is associated with cancers of the cervix, vulva, vagina, anus, penis and the oropharynx [100]. In immunosuppressed individuals, including PLWH, HPV infection often becomes chronic [101]. In particular, PLWH have a higher incidence of HPV infection, abnormal pap smears and persistent HPV infection due to the less efficient viral clearance, leading to a high risk of HPV-related cancers [101].

Because of the more common and persistent HPV-related complications in PLWH, HPV vaccination programs are encouraged in this population. HPV vaccination represents the main preventive tool for HPV-related cancers as well as other HPV-related diseases [102]. HPV vaccines are based on the recombinant protein virus-like particle (L1 VLP) with a proprietary adjuvant. Currently, there are three licensed prophylactic L1 VLP-based vaccines that provide protection against two (bivalent, commercialized in 2008), four (quadrivalent) and nine (nonavalent) HPV genotypes [103]. Given the higher vulnerability of PLWH, particularly in men who have sex with men (MSM) who are also PLWH, to acquire multisite infections mostly characterized by various genotype combinations and the ability of the nonavalent vaccine to prevent 80% of HPV infections, vaccination programs with this nine-genotype protection should be implemented, especially among MSM [104,105].

HPV vaccines have proved their safety, efficacy and effectiveness in immunocompetent young persons, leading to a standard vaccination regime for young girls and boys (aged 9–14 years) reduced from the originally licensed three-dose regimen to two doses [106,107].

Serum neutralizing antibodies are thought to be the major protective branch of adaptative immunity afforded by L1 VLP-based vaccines, although CD4+ T cells are involved in the induction and re(activation) of antigen-specific memory B cells, leading to high antibody levels and, therefore, are critical for long-term vaccine-induced protection [108].

The immune response elicited by the quadrivalent HPV vaccine seems to persist in vaccinated individuals up to 5 years post-vaccination [109].

Concerning PLWH, licensed HPV vaccines have proven to be generally safe and well tolerated [110]. Lower rates of antibody levels elicited by similar vaccine constructs have been observed, raising concern about the efficacy of HPV vaccines in PLWH [111]. However, several studies have reported that the immune response induced in PLWH is similar to that found in general population, with high rates of seroconversion and a cellular immunogenicity comparable to that of general population [98,110,112,113,114]. Furthermore, antibody levels following vaccination appear to be stable over time [115].

Significant positive correlations between T-cell responses and current absolute CD4+ T-cell count together with negative correlations between such responses and HIV viremia have been observed [116,117]. Furthermore, higher seroconversion rates among PLWH with current absolute CD4+ T-cell counts >200 cells/μL compared with ≤200 cells/μL have also been reported [116,117]. However, a possible decline in B-cell memory responses between 2 and 5 years after the last vaccination dose has been described in PLWH [116]. This is consistent with observations focused on the characterization of the immunogenicity of other vaccines, but few immunogenicity studies of HPV vaccines among PLWH include participants with low absolute CD4+ T-cell counts, supporting the need to further elucidate their immune capacity. In addition, to the best of our knowledge, few studies are focused on the immunogenicity of the nonavalent HPV vaccine, and information is lacking about the quality of such response.

## 5. Influenza Vaccination

Influenza is an infectious respiratory disease with annual estimations of approximately 1 billion infections, 3–5 million cases of severe illness and 300,000–500,000 deaths, according to WHO [118]. Influenza viruses belong to the *Orthomyxoviridae* family and account for three different types: influenza A, B and C. All three types share common characteristics, such as the segmented genome, made of negative-sense single-stranded RNA, and the presence of an envelope (derived from host cell membrane) with glycoproteins, essential for viral entry in target cells [119]. Viral particles are enveloped, and surface glycoproteins hemagglutinin (HA) and neuraminidase (NA) represent the major antigenic determinants [120]. Influenza viruses are characterized by antigenic variation, based on two different mechanisms: antigenic drift (present in all influenza types) and antigenic shift (characteristic of influenza A only) [120].

Influenza B and C have a narrower host range (humans only and humans and swine, respectively) than influenza A, which can infect humans, swine, equine, other mammals such as ferrets, felids, mink, dogs, civets, marine mammals and avians [119]. Influenza A and B viruses are most relevant clinically, since they cause severe respiratory infections in humans [121]. Sequencing has confirmed that these viruses share a common genetic ancestry. They have genetically diverged, and an exchange of viral RNA segments between viruses has been reported occurring within each genus or type, but not across types [120]. Influenza A viruses are further characterized by the subtype of their surface glycoproteins, the HA and the NA. There are 18 different HA subtypes and 11 different NA subtypes [122]. Unlike influenza A, influenza B is not further divided into subtypes [123].

During infection, the epithelial cells are the primary targets for influenza viruses. These cells line the respiratory tract and initiate an antiviral immune response upon influenza virus detection. The first line of defense is represented by the innate immune system and constituted by physical barriers and the innate immune cellular responses [124]. A critical role is performed by the adaptive immunity in the clearance of viral pathogens during the later stages of infection. Furthermore, respiratory mucosal immunity is induced in the related mucosal tissues during influenza infection and involved in antiviral defense [125].

Generally, infections occur in children, although most of the severe cases involve very young or elderly individuals and individuals affected by chronic pulmonary or cardiac conditions, diabetes mellitus or immunocompromising conditions [126]. Considering the periodical recurrence of influenza infection and the severe complications occurring in the elderly and in patients with concomitant chronic diseases, the influenza vaccine represents an essential tool for preventing infection and limiting the burden of the disease. The choice of relevant antigens remains of paramount importance in developing influenza vaccines which are formulated every year to match the circulating strains.

Currently, there are three kinds of vaccines, inactivated, live attenuated and recombinant HA vaccines, licensed in different countries [127]. The WHO recommends seasonal influenza vaccination to children 6 months to 5 years of age, elderly individuals (>65), all persons with chronic medical conditions and pregnant women. All available influenza virus vaccines are injected intramuscularly, except for the live attenuated influenza virus vaccines, which are administered intranasally [128].

Debate is still ongoing on the efficacy and effectiveness of the licensed vaccines, although most studies find a positive effect of vaccination on vaccinated individuals [129]. The effectiveness of influenza vaccines has been found to be related predominantly to the age and immune competence of the vaccinated individual and the antigenic relatedness of vaccine strains to circulating strains [130].

Currently licensed influenza vaccines focus on the production of antibodies against the viral HA, which binds host receptors to mediate viral entry, neutralizing the virus and preventing the infection [131]. However, the decline in vaccine-specific antibodies and the antigenic drift of influenza viruses over time leads to the necessity of annual revaccination. Targeting T-cell responses seems to be a promising technique to ameliorate influenza vaccines, although it does not prevent infection, but it can reduce the severity of the infection [131].

Indeed, the role of T-cell immunity was demonstrated during the 2001 H1N1 pandemic, where the magnitude of the pre-existing cytotoxic T-lymphocyte (CTL) response inversely correlated with disease severity in individuals without detectable neutralizing antibody [132].

PLWH experience prolonged duration of influenza infection and increased severity of illness, together with higher rates of hospitalizations compared to the general population [133]. As a result, annual vaccination against seasonal influenza is recommended by many national immunization guidelines [134].

Again, the success of influenza vaccination is related to current absolute CD4+ T-cell count [135]. Indeed, weaker response rates were observed in PLWH with lower current absolute CD4+ T-cell count, probably due to impaired function of the peripheral blood Tfh and B-cell functions [136,137].

However, the data are not always in accordance, as demonstrated by the study conducted by Tebas et al. [138], aimed at evaluating safety and immunogenicity of the H1N1 2009 vaccine in PLWH. The authors showed that only 60% of the participants developed protective antibody titers after immunization [138]. On the contrary, a clinical trial (P1088) launched by the International Maternal Pediatric and Adolescent Clinical Trials (IMPAACT) Network evaluated safety and efficacy of a monovalent pandemic H1N1 (pH1N1) vaccine in perinatally HIV-1-infected children and adolescents, showing that two doses of double-strength pH1N1 vaccine are safe and immunogenic and may provide improved protection against influenza in this population [139].

Concerns remain about the efficacy in elderly PLWH, as observed in the general population. Alternative vaccines, dosing, adjuvants or schedule strategies may be needed to achieve effective immunization of this vulnerable population.

## 6. Monkeypox Virus Vaccination

Monkeypox virus infection (MPXV), also commonly known as “monkey pox” or simian orthopoxvirus, is an infectious disease caused by an *Orthopoxvirus* (family *Poxviridae*). There are two genetically distinct MPXV clades that exhibit different lethality rates. Clade II comprising the first cases of infections was reported in West Africa and clade I in Central Africa. Clade IIb was responsible for the global epidemic outbreak in May 2022. Monkeypox virus infection presents a clinical picture that can vary according to the clades: clades I and IIa resemble smallpox and clade IIb is characterized by atypical presentations. During the 2022 epidemic which raged in countries where the disease was not endemic, the symptoms were very polymorphic (cutaneous and mucous membrane involvement, painful lymphadenopathy, angina, anitis or proctitis, etc.) which could lead to more complicated forms (ocular involvement, encephalitis or encephalopathies, multiorgan involvement in otherwise immunocompromised patients). Transmission to humans occurs from an animal reservoir or from human to human via direct or indirect physical contact (contaminated objects).

Despite the announcement of the end of the MPXV epidemic by the WHO in May 2023 [140], questions around MPXV remain topical in the scientific community, especially in immunocompromised subjects such as PLWH who are considered as a population at risk [141]. Nearly 957 cases of monkeypox virus (MPXV) infection in Italy and 25,887 cases of infection in Europe have been reported so far [142].

Recent studies have described the fatal nature of the MPXV infection in a subpopulation of these PLWH characterized by absolute CD4+ T-cell counts of >200 cells/µL, presenting a clinical picture marked by massive necrotizing skin and cutaneous, genital and non-genital mucosal lesions, which can sometimes be accompanied by pulmonary involvement with multifocal nodular opacities or respiratory failure and severe cutaneous and blood bacterial sequelae which, in 15% of cases, led to death [141,143,144].

Most cases of MPXV infections reported in Europe and North America since May 2022 were mainly transmitted among men who have sex with men (MSM) with evidence of an increased prevalence of HIV and other sexually transmitted infections (STIs). Given the morbidity and lethality in PLWH, a strong evolution of the therapeutic arsenal against MPXV with many vaccines has been made available to stem the epidemic [145].

The third-generation vaccine contains the live modified attenuated virus of vaccinia Ankara. MVA-BN is currently the only approved vaccine in areas where sufficient vaccine stocks are available. Jynneos/MVA-BN is used for pre-exposure prophylaxis to MPXV in HIV-infected individuals; it is also indicated for the prevention of MPXV in individuals 18 years of age and older who are at high risk of infection [146,147].

To date, there are very few data on an immune response of the MVA-BN vaccine against MPXV in PLWH. In a study that focused on evaluating the safety and immunogenicity of MVA-BN in immunocompromised subjects [148], a phase II trial was conducted between 2006 and 2009 in the United States and Puerto Rico; a total of 579 volunteers were recruited into the study: 439 vaccine-naïve subjects (88 immunocompetent subjects, 351 PLWH) and 140 vaccine-experienced subjects (9 immunocompetent subjects, 131 PLWH) received at least one vaccination. The results of this study demonstrated that the MVA-BN vaccine presents a better safety and tolerance profile in PLWH with absolute CD4+ T-cell counts < 200 cells/µL than in immunocompetent subjects, regardless of their previous smallpox vaccination status. In other subsequent studies, the safety profile of the MVA-BN vaccine in immunocompromised subjects, particularly those infected with HIV, was comparable or even better in terms of local reactions than in subjects not infected with HIV [149,150].

The results of the studies showed that the safety profile of the MVA-BN vaccine in immunocompromised subjects, particularly PLWH, considered at risk for conventional smallpox vaccination, was comparable or even better in terms of local reactions than in the general population [148,150]. Antibody responses were also comparable between immunocompetent subjects and PLWH.

## 7. SARS-CoV-2 Vaccination

With more than 757 million confirmed cases, the coronavirus disease 2019 (COVID-19), caused by severe acute respiratory syndrome coronavirus 2 (SARS-CoV-2), is the third coronavirus disease in the past 20 years [151,152].

SARS-CoV-2 is an enveloped, positive-sense, single-stranded RNA virus that belongs to the *Coronaviridae* family [153]. Its genome encodes for four major structural proteins, namely the spike surface glycoprotein (S), which is responsible for the binding to the host receptor *angiotensin-converting enzyme 2* (ACE2), the small envelope protein (E), the matrix protein (M) and the nucleocapsid protein (N), and other non-structural proteins [154]. Viral transmission can occur by direct, indirect or close contact by infected people through secretions (saliva or respiratory droplets) [155]. SARS-CoV-2 infects bronchial epithelial cells, pneumocytes and upper respiratory tract cells in humans, developing into severe, life-threatening respiratory diseases and lung injuries [156].

Many countries have launched vaccination campaigns to prevent SARS-CoV-2 infection, and several vaccines have been approved by the World Health Organization (WHO), although many obstacles to global vaccination remain [157]. Among the vaccines based on different technologies that have been developed during the health emergency, messenger RNA (mRNA)-based vaccines have been widely used to contain the pandemic [158].

In the general population, mRNA-based vaccines have proven to elicit a robust and protective humoral and cellular response against the SARS-CoV-2 S protein, reducing mortality and morbidity related to SARS-CoV-2 infection [159,160,161]. In addition, the specific T-cell response induced by the vaccine, with the ability to recognize different regions of the S protein, contributes to the vaccine’s effectiveness against viral variants and protects individuals from severe forms of COVID-19 [162,163].

It has been estimated that PLWH represent 1% of total hospitalized cases and, differently from HIV infection which, in the absence of ART, is invariably fatal, COVID-19 disease is highly variable, ranging from mild to severe and critical forms of illness [164,165].

During the health emergency, vaccination of PLWH became of vital significance and strongly recommended by health authorities because of the potentially worse outcomes after SARS-CoV-2 infection, although reports about the increased risk of severe COVID-19 in this population are in some cases contradictory [166]. However, PLWH may experience a higher burden of various comorbidities, many of which have emerged as risk factors for severe COVID-19. Key risk factors for severe COVID-19 include both non-HIV comorbidities known to be associated with severe disease like older age, diabetes, obesity and cardiovascular disease as well as HIV-specific risk factors such as low absolute CD4+ T-cell count, viremia and *Mycobacterium tuberculosis* co-infection [167]. Furthermore, the suboptimal responses to other vaccines have raised concerns about the efficacy of vaccines against SARS-CoV-2 in this potentially more vulnerable population.

Some HIV viral blips following mRNA vaccinations have been reported; however, licensed vaccines have proven to be safe and efficacious in PLWH with stable absolute CD4+ T-cell counts and well-controlled viremia [168].

In particular, published data on the immunogenicity of mRNA vaccines show values of anti-S antibodies, neutralizing antibody activity and cellular immune responses in PLWH on ART and with current absolute CD4+ T-cell counts above 200 cells/µL comparable to those observed in the general population after a primary vaccination cycle [168,169,170]. Such a response was found to be significantly inferior in PLWH with current absolute CD4+ T-cell count < 200 cells/µL compared to those with >500 cell/mm^3^ and the general population, suggesting that the immunogenicity at the time of vaccination is related to the current absolute CD4+ T-cell count [171,172].

In September 2021, the administration of an additional booster of anti-SARS-CoV-2 mRNA vaccine was approved in Italy to be given after >28 days after completion of the primary vaccination cycle in PLWH depending on current absolute CD4+ T-cell count and/or detectable HIV viremia [173]. The third dose improved the responsiveness particularly in PLWH on ART with current absolute CD4+ T-cell counts < 200 cells/μL, improving both the rate and the magnitude of the response and supporting the additional dose strategy in this category of patients with severe immune impairment [172,174].

Some strategies aimed at increasing the tolerability of the mRNA vaccine, such as the use of pidotimod, which was able to reduce vaccination-related adverse events, could be useful to encourage people to received vaccination [175].

In PLWH with a current absolute CD4+ T-cell count > 200 cells/μL, T-cell response elicited by the third dose was like that induced by the primary vaccination cycle, suggesting that the first two doses were able to achieve full T-cell immunization. Furthermore, the increased humoral response is consistent with the hypothesis that the third dose is able to induce a robust B-cell memory response, previously elicited by the primary vaccination series [176].

However, questions remain about mRNA vaccine’s immunogenicity in PLWH with ongoing immunosuppression and viremia who represent a particularly vulnerable group that is poorly represented in vaccine trials. Furthermore, recent studies have shown a lower polyfunctional capacity in this population, as already described in the setting of other co-infections, raising issues about the real capability of their immune response [174,177].

## 8. Streptococcus Pneumoniae Vaccination

*Streptococcus pneumoniae* (*S. pneumoniae*), a Gram-positive bacterium, is the most significant cause of bacterial disease in humans. A variety of clinical syndromes are related to its infection, including pneumonia, meningitis, bacteremia, acute otitis media and sinusitis [178]. Despite the availability of a broad arsenal of antibiotics and a vaccine, worldwide, approximately 14.5 million cases of serious pneumococcal diseases per year have been reported, leading to approximately 826,000 deaths [179].

In PLWH, invasive pneumococcal disease (IPD) and pneumococcal pneumonia continue to pose a challenge with high recurrence rates [180], significant public health impact, morbidity and a high mortality rate of up to 25% [181].

The introduction of pneumococcal vaccines has significantly reduced morbidity, although PLWH still remain at a 30-times-higher risk of IPD as compared to the general population [182]. Specifically, an absolute CD4+ T-cell count < 200 cells/µL and high levels of HIV RNA have been strongly associated with the risk of IPD [182].

Four pneumococcal vaccines are currently available: PCV13, PPSV23, PCV15 and PCV20 [183]. The PCV13 contains protein-conjugated polysaccharides of 13 serotypes of pneumococci [184]. The PCV15 and PCV20 contain all the PCV13 serotypes, with two additional serotypes in PCV15 and seven additional serotypes in PCV20 [183]. The PPSV23 contains 23-valent pneumococcal polysaccharides [185]. The PCV15 is administered as a single dose with one PPSV23 follow-up dose given at least 8 weeks later; no additional doses are recommended after that. The PCV20 requires one dose only; there are no additional doses needed [183].

There are limited data on the efficacy of pneumococcal vaccination in PLWH. Retrospective studies indicate that PPSV23 alone has modest clinical benefit, if any, in reducing rates of pneumococcal infections [186,187]. The immune response induced by PPSV23 is a T-cell-independent humoral response, while PCV13 induces a T-cell-dependent response producing pneumococcal serotype-specific antibodies and memory B cells which provide long-lasting protection [188]. Sequential vaccination with PCV 13 followed by PPSV23 provides a prime boost effect on inducing and maintaining protective immunity [189].

These vaccines are less immunogenic in PLWH compared to the general population due to a mitigated immune response. The combination of PPSV23 and PCV 13 has been shown to be more immunogenic than either of the vaccines alone and is recommended internationally for prevention of IPD in PLWH [190]. Moreover, other strategies to improve the immunogenicity of pneumococcal vaccines in PLWH were performed. Indeed, in a double-blind, placebo-controlled study, the addition of adjuvant CPG 7909, a toll-like receptor agonist, significantly enhanced the proportion of high responders to the vaccine [191].

According to the Italian Vaccination Plan (2017–2019), the tetanus, diphtheria and pertussis (Tdap) vaccine is co-administered with PCV and HBV vaccines. For PLWH of ≥11 years old who have never received any vaccine, three doses of Tdap are administered, with an interval of 0, 1 month, 6–12 months. In individuals with advanced-stage HIV, the response is suboptimal for both tetanus and diphtheria, while in subjects with CD4 > 300 cells/µL, the response against tetanus is optimal, comparable to subjects without HIV while, for diphtheria, it can remain markedly lower [192].

## 9. Varicella Zoster Virus Vaccination

The Varicella zoster virus (VZV), a double-stranded DNA ubiquitous human alphaherpesvirus [193], causes varicella, establishes lifelong latency in ganglionic neurons and reactivates later in life to cause herpes zoster, commonly associated with chronic pain [193,194,195,196].

Varicella and herpes zoster are more common and more severe in the elderly, the female sex [197] and in people who are immunocompromised, such as PLWH and people taking immunosuppressive drugs and chemotherapy [198]. The incidence of herpes zoster is more than 15 times higher in PLWH compared to age-matched immunocompetent subjects. Herpes zoster can occur in PLWH at any absolute CD4+ T-cell count, but disease frequency is highest when absolute CD4+ T-cell counts are below 200 cells/µL [199,200,201].

Despite the mandatory vaccination of children against chickenpox in the early 1995s in the United States [202] and from 2003 in Europe [203] (this led to immunization (83% to 95%) of the general population [204]), the risk of VZV reactivation remains particularly high in seropositive adults [205,206]. The incidence of herpes zoster is approximately 4–7 cases/1000 person-years and without vaccination, the lifetime herpes zoster risk is 20–30% [207,208].

Vaccination offers an option that could overcome the challenges associated with conventional antiviral prophylaxis while potentially providing longer-lasting protection against shingles [209]. The live zoster vaccine (ZVL) is a live attenuated vaccine approved for people aged ≥60 years [210]. The effectiveness of the ZVL vaccine may decrease with age and it is generally contraindicated in immunocompromised subjects due to its potential infection risks [211,212].

Recently, the recombinant zoster vaccine (RZV), an adjuvanted subunit vaccine recommended for use in adults ≥ 50 years of age since 2017 by the Advisory Committee on Immunization Practices (ACIP) [213], was also approved by the ACIP for the prevention of herpes zoster in adults aged 19 and older who have or will have an increased risk of shingles due to immunodeficiency or immunosuppression caused by diseases or treatment [210].

The efficacy of RZV in immunocompromised subjects is lower than in immunocompetent subjects, reflecting cell-mediated immunodeficiency and a weaker immune response due to an underlying immunocompromised state [214]. Clinical trial data comparing memory T-cell responses to both vaccines mentioned (ZVL and RZV) found higher responses in RZV recipients, and only RZV recipients had five-year persistence of higher responses [215,216]. The efficacy of RZV is high, even in people aged ≥70 years [217]. Pooled analyses also showed that the vaccine was 91.3% effective against shingles in participants over the age of 70 [218,219]. The clinical efficacy of the RZV vaccine has also been demonstrated in various phase II and III, placebo-controlled, observer-blinded studies conducted in immunocompromised adults aged 18 years and older with two doses administered 1–2 months apart [220,221].

Regarding the safety of the RZV vaccine, results from an observational study showed no difference between immunocompetent and immunocompromised groups, indicating that immunosuppression may not be a determinant of adverse vaccine effects [221].

These preliminary data confirm the efficacy conferred by the RZV vaccine against herpes zoster. However, these data should be interpreted with caution and require in-depth studies.

## 10. Conclusions

Despite ART-induced virologic suppression, PLWH remain at increased risk of mortality and morbidity from vaccine-preventable diseases, in part because of persistent immunopathology, resulting in a compromised response to vaccination, and vaccine-induced antibodies may fade more rapidly in PLWH than in the general population [1,16].

Moreover, besides the primary response, long-term persistence of protection has been poorly documented and recommendations on the timing of booster injections are based on data collected in the general population, although patterns of antibody decay may differ. In this regard, it is necessary to estimate how seroprotection declines over time among patients who initially responded to immunization.

Many efforts have been made during the SARS-CoV-2 pandemic to evaluate vaccine efficacy in PLWH. The findings obtained further confirm the critical role of CD4+ T cells as a key factor of effective humoral responses and predictor of vaccine success. In addition, evidence about the complementary role of T cell-specific responses in mediating protection has emerged, particularly in individuals with low seroconversion rates, reducing mortality and morbidity related to SARS-CoV-2 infection. However, what constitutes protective immunity is still discussed, making it difficult to define protective efficacy of vaccines. In determining vaccine scheduling and efficacy, CD4+ T-cell count, CD4/CD8 ratio and viremia should be considered, with the awareness that it will not capture the full immune profile of this population.

In fact, it is becoming increasingly clear that PLWH represent a diverse population in terms of immune phenotype, with the consequence that different subgroups require different vaccination strategies to improve their immunological responses.

Furthermore, the setting of co-infection poses additional concerns, particularly regarding T-cell immunity, since with the intersecting of SARS-CoV-2, HIV and TB epidemics, SARS-CoV-2-specific CD4+ T cells have shown a lower polyfunctional capacity.

In our opinion, a better understanding of these issues will help guide vaccination and prevention strategies for PLWH.

We should also consider that male adults living in Europe and in the United States are the most represented participants in the studies, which poorly reflects the global prevalence of PLWH, and that, with the pandemic, a reduction in the access to ART and in vaccine coverage may leave PLWH potentially more vulnerable.

To date, studies assessing long-term immunogenicity, planned with scientific rigor, are needed. An improvement in the field of vaccine development could bring changes in the lives of PLWH. In conclusion, the main preventive tool for many infectious diseases remains vaccination, together with counseling and screening programs. However, greater attention needs to be paid to PLWH with uncontrolled viral infection and/or low CD4+ T-cell counts and to the effects of aging and comorbidities.

## Figures and Tables

**Figure 1 viruses-15-01844-f001:**
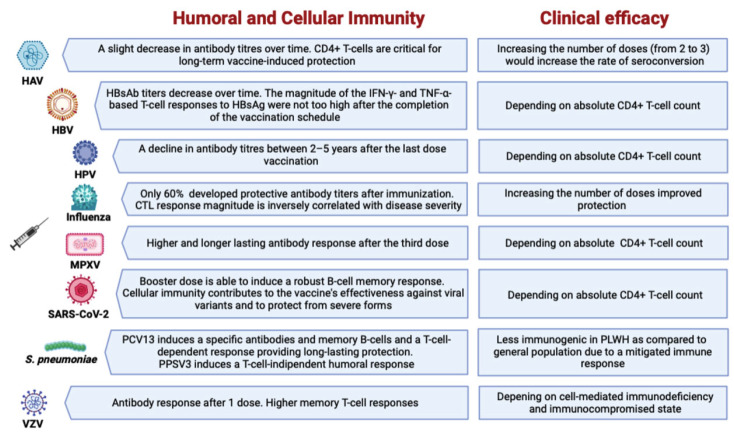
Summary of humoral and cellular immunity in PLWH to recommended vaccinations.

**Table 1 viruses-15-01844-t001:** Recommended PLWH Immunization.

Pathogen	Vaccine Platform	Absolute CD4+ T Cell Count	
		<200	>200
HAV	Inactivated	2–3 doses (varies by formulation)	
HBV	Recombinant	2–4 doses (varies by formulation)	
HPV	Recombinant	3 doses through age 26	
Influenza	Inactivated (IIV)	1 dose annually	
	Recombinant (RIV)	1 dose annually	
	Live, attenuated (LAIV)	not recommended	
MPXV	Live, attenuated	not recommended	2 doses
SARS-CoV-2	mRNA-based	2 doses + booster	
	Viral vector	2 doses	
	Recombinant	2 doses	
*Streptococcus pneumoniae*	PCV15 PCV20 PPSV23	1 dose PCV15 followed ≥8 weeks by 1 dose PPSV23 or 1 dose PCV20	
VZV	Live, attenuated (ZVL)	not recommended	
	Recombinant (RZV)	2 doses for 18 and older	

HAV: Hepatitis A virus, HBV: Hepatitis B virus, HPV: Human Papilloma virus, MPXV: Monkeypox virus, VZV: Varicella zoster virus, SARS-CoV-2: Severe acute respiratory syndrome coronavirus 2, PCV: pneumococcal protein-conjugated vaccine, PPSV: pneumococcal polysaccharide vaccine.

## Data Availability

Not applicable.

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
