# Peer review of "Immunogenicity and Efficacy of Vaccination in People Living with Human Immunodeficiency Virus"

_viruses, 2023, doi:10.3390/v15091844_

Round 1

Reviewer 1 Report

The paper analyzes the immunogenicity and efficacy of vaccinations in the HIV positive population. The review appears  is well done and extensive, however, in my opinion, some changes are needed before publication.

The paragraph on HPV, focuses mainly on 4-valent vaccination. The authors could discuss  by the different HPV strains circulation  in the HIV population, especially MSM (10.4102/sajid.v37i1.363 ,10.3892/ol.2017.7140, 10.1002/jmv.24943) and therefore on the greater coverage offered by  9-valent vaccines.

 In the paragraph on sars cov 2 vaccination, the authors could talk about strategies to improve vaccine efficacy (10.4084/MJHID.2022.023.)

it is not understood why the paragraph on pnuemococcus vaccination was named 7.1 and not 8. It be useful 

The authors talk about  the low immunogenicity of the vaccine but do not cite any paper aimed at increasing its efficacy (ex. 10.3390/nu13124412.)

the authors talk about the immunological activation in hiv patients but do not mention pharmacological strategies aimed at improving the immune system  ( ex. 10.2174/1570162X18666210111102046)

Author Response

Reviewer 1

The paper analyzes the immunogenicity and efficacy of vaccinations in the HIV positive population. The review appears well done and extensive, however, in my opinion, some changes are needed before publication.

The paragraph on HPV, focuses mainly on 4-valent vaccination. The authors could discuss by the different HPV strains circulation in the HIV population, especially MSM (10.4102/sajid.v37i1.363 ,10.3892/ol.2017.7140, 10.1002/jmv.24943) and therefore on the greater coverage offered by  9-valent vaccines.

As suggested by the Reviewer, we implemented the paragraph on HPV adding two articles (Ucciferri, C.; Tamburro, M.; Falasca, K.; Sammarco, M.L.; Ripabelli, G.; Vecchiet, J. Prevalence of Anal, Oral, Penile and Urethral Human Papillomavirus in HIV Infected and HIV Uninfected Men Who Have Sex with Men. J. Med. Virol. 2018, 90, 358–366, doi:10.1002/jmv.24943 and Tartaglia, E.; Falasca, K.; Vecchiet, J.; Sabusco, G.P.; Picciano, G.; Di Marco, R.; Ucciferri, C. Prevalence of HPV Infection among HIV‑positive and HIV‑negative Women in Central/Eastern Italy: Strategies of Prevention. Oncol. Lett. 2017, 14, 7629–7635, doi:10.3892/ol.2017.7140.) about the higher vulnerability of PLWH to acquire infections characterized by different HPV strains and therefore the utility of 9-valent vaccines thanks to their higher coverage against different HPV genotypes (lines 237-241).

 In the paragraph on SARS-CoV-2 vaccination, the authors could talk about strategies to improve vaccine efficacy (10.4084/MJHID.2022.023.)

In the paragraph about SARS-CoV-2 vaccination, we only mentioned the “additional dose strategy” to improve vaccine efficacy because we focused our attention on mRNA-based vaccines, that don’t contain adjuvants. However, as kindly suggested by the Reviewer, we added an article (Ucciferri, C.; Vecchiet, J.; Auricchio, A.; Falasca, K. Improving BNT162b2 MRNA Vaccine Tolerability without Efficacy Loss by Pidotimod Supplementation. Mediterr. J. Hematol. Infect. Dis. 2022, 14, e2022023, doi:10.4084/MJHID.2022.023) about the use of pidotimod, an immunomodulatory molecule that improved mRNA vaccine tolerability without altering the immune response to the vaccine (lines 461-463).

it is not understood why the paragraph on pnuemococcus vaccination was named 7.1 and not 8. It be useful...

As noted by the Reviewer, we changed the name of the paragraph on Pneumococcus vaccination in 8 instead of 7.1.

The authors talk about the low immunogenicity of the vaccine but do not cite any paper aimed at increasing its efficacy (ex. 10.3390/nu13124412.)

In lines 506-511, in Paragraph 8, we reported some strategies aimed at increasing the efficacy of pneumococcal vaccination, such us the use of the combination of PPSV23 and PCV 13, that has been shown to be more immunogenic than either of the vaccines alone, and therefore recommended internationally for the prevention of IPD in PLWH, and we cited a double-blind, placebo-controlled study, that evaluated the effect of the addition of an adjuvant CPG 7909, a toll-like receptor agonist, that significantly enhanced the proportion of vaccine high responders.

the authors talk about the immunological activation in hiv patients but do not mention pharmacological strategies aimed at improving the immune system (ex. 10.2174/1570162X18666210111102046)

As suggested by the Reviewer, we added an article about the use of immunomodulatory molecules that have beneficial effects on the residual immune activation in PLWH (Ucciferri, C.; Falasca, K.; Reale, M.; Tamburro, M.; Auricchio, A.; Vignale, F.; Vecchiet, J. Pidotimod and Immunological Activation in Individuals Infected with HIV. Curr. HIV Res. 2021, 19, 260–268, doi:10.2174/1570162X18666210111102046.) (lines 44-45).

Reviewer 2 Report

The review article was written nicely. In this article authors described various vaccinations programs and the immunogenic effect of these vaccines in PLWH. A common vaccine, Tdap, which is generally given to children and pregnant mother for diphtheria, tetanus, and whooping cough. I think Tdap might have some immunogenic efficacy with PLWH, especially for pregnant mothers. If the authors would highlight this area would be better. Another vaccine for tuberculosis (BCG) would have some effect with PLWH. Otherwise, this is a nice review article. 

Author Response

Reviewer 2

The review article was written nicely. In this article authors described various vaccinations programs and the immunogenic effect of these vaccines in PLWH. A common vaccine, Tdap, which is generally given to children and pregnant mother for diphtheria, tetanus, and whooping cough. I think Tdap might have some immunogenic efficacy with PLWH, especially for pregnant mothers. If the authors would highlight this area would be better. Another vaccine for tuberculosis (BCG) would have some effect with PLWH. Otherwise, this is a nice review article. 

As kindly suggested by the Reviewer, we implemented the review including some information about Tdap in PLWH (lines 512-518), while considerations about BCG vaccine in PLWH were already reported in lines 54-56.

Round 2

Reviewer 1 Report

The authors responded well to the objections raised. The paper can be accepted in the present form